# An Experimental and Theoretical Determination of Oscillatory Shear-Induced Crystallization Processes in Viscoelastic Photonic Crystal Media

**DOI:** 10.3390/ma14185298

**Published:** 2021-09-14

**Authors:** Chris E. Finlayson, Giselle Rosetta, Jeremy J. Baumberg

**Affiliations:** 1Department of Physics, Prifysgol Aberystwyth University, Aberystwyth SY23 3BZ, UK; grm4@aber.ac.uk; 2Cavendish Laboratory, Department of Physics, University of Cambridge, Cambridge CB3 0HE, UK

**Keywords:** polymers, shear-induced crystallization, photonic crystals, composite materials, viscoelasticity

## Abstract

A study is presented of the oscillatory shear-ordering dynamics of viscoelastic photonic crystal media, using an optical shear cell. The hard-sphere/“sticky”-shell design of these polymeric composite particles produces athermal, quasi-solid rubbery media, with a characteristic viscoelastic ensemble response to applied shear. Monotonic crystallization processes, as directly measured by the photonic stopband transmission, are tracked as a function of strain amplitude, oscillation frequency, and temperature. A complementary generic spatio-temporal model is developed of crystallization due to shear-dependent interlayer viscosity, giving propagating crystalline fronts with increasing applied strain, and a gradual transition from interparticle disorder to order. The introduction of a competing shear-induced flow degradation process, dependent on the global shear rate, gives solutions with both amplitude and frequency dependence. The extracted crystallization timescales show parametric trends which are in good qualitative agreement with experimental observations.

## 1. Introduction

Iridescent 3D photonic structures with systematic structural ordering can be found in opal gemstones, and in many other manifestations in nature [1,2,3,4]. These are microstructures with a wavelength-scale dielectric periodicity, with an inherent ability to give distinguishing optical properties (e.g., structural color), which are not accessible in a comparable fashion using dyes or pigments [5,6,7,8]. Whilst methods such as holography or imprinting enable these effects to be replicated to some extent on 2D surfaces, genuine 3D bulk structures have generally been more challenging to engineer artificially. This is particularly the case in striving for large scale assembly methods, that are sufficiently cost effective to facilitate widespread application. Widely studied strategies for assembling bulk-ordered optical materials have conventionally relied upon the self-assembly of high and low refractive index components [9,10,11,12,13,14,15]. However, the resultant structures lack the mechanical tractability and robustness needed for many practical applications and, critically, any reproducible bulk-scaling remains very limited. 

By marked contrast, the authors’ recent work on “polymer opals” (POs), based on arrays of composite polymer microparticles, has demonstrated how such synthetic opals are an archetypal platform for next generation bulk-scale photonic crystals, coatings, and smart materials [16,17,18,19,20,21,22,23,24]. These mass-produced particles, constructed of rigid polystyrene sphere cores with a grated-on softer ethyl-acrylate shell, may be permanently shear-assembled into permanent solvent-free quasi-solids.

The design of these polymeric composite particles in illustrated in Figure 1; the hard-sphere/“sticky”-shell produces a rubbery bulk medium, with a characteristic viscoelastic rheology. Control of particle diameters over the range of around 200–300 nm allows tuning of the Bragg wavelength, and associated vibrant structural color, over the whole visible and near infrared spectral region. The Bending-Induced Oscillatory Shear (BIOS) crystalline ordering process is also still possible with polydispersity levels far beyond that feasible for colloidal self-assembly, thus greatly reducing the requirements for low particle size dispersity [25,26]. Both BIOS and the related edge-induced rotational shearing (EIRS) process [16] can thus reproducibly generate 3D opals over areas of square-meters and film thicknesses of several hundred microns; a system which may thus be considered to be the largest nano-assembled ordered structures ever demonstrated [20,27,28].

In BIOS, strong ordering forces within the films are generated by lateral shearing of the disordered melt of nanoparticles, leading to the formation of close-packed solvent free periodic nanostructures. Core–shell precursor spheres are homogenized by extrusion and then rolled into thin films laminated between two rigid PET sheets. The BIOS process is then applied to the sandwich structure, and an ordered PO layer is obtained. The BIOS processing cycle is achieved by mechanically oscillating the sandwich structure around a fixed cylindrical surface under tension at a stabilized temperature. This generates strong shearing forces inside the PO purely parallel to the surface and resultant strains of magnitude up to 300%. Multiple reported crystallographic and microscopic characterizations [17,29,30] have repeatedly confirmed a random hexagonal packing arrangement, with some in-plane layering, and a progressive development of ordering through the structure from the surfaces. The final PO thin films show exceptional mechanical robustness, flexibility, and stretchability (>100%), allowing for the tuning of optical properties by viscoelastic deformation [19,31]. 

Whilst shear-induced ordering methods in POs have been demonstrated in detail and the end products characterized, relatively little was known concerning the time dependence of photonic crystal formation or the underlying microscopic mechanisms, until the direct measurement of monotonic ordering dynamics in a shear-cell geometry by Snoswell et al. [32]. Certainly, no complete theoretical understanding or models of this ordering yet exist, despite the evident utility in facilitating a host of new scientific insights into tunable analog structures, which are mechanically impossible in more conventional “monolithic” photonic structures.

We might make an instructive comparison to, and also draw pertinent distinctions with, some other systems of shear assembly of micron-sized particles, such as low viscosity colloidal suspensions [33,34,35]. Recent studies have described the effects of oscillatory shear under a range of conditions in such systems [36,37,38,39,40,41,42], where continuous shear can crystalize colloidal monodisperse particles when there is a suitable fluid medium present. A notable variant of these methods is shear alignment in anisotropic ensembles [43,44]. As a fundamental distinction, POs do not contain a discrete fluid phase, and therefore cannot be directly compared to these systems of colloidal suspensions. Rheological studies [45,46] have demonstrated that the grafted soft-shell polymer forms a quasi-continuous viscoelastic matrix during the shear-ordering process, and mobility of the (athermal, nondiffusive) rigid spheres is highly inhibited by the gum-like medium. There is strong viscoelastic dissipation inside the system, and the characteristic Péclet number, *Pe* = $\dot{\gamma }$*a*^2^/*D*_0_ (where *a* is the particle radius,
$\dot{\gamma }$
the shear rate, and *D*_0_ the Stokes-Einstein diffusion coefficient) [47] in the opals is therefore consistently many orders of magnitude greater than in the colloidal suspensions. For typical values of
$\dot{\gamma }$
~ 1 s^−1^, *a* ~ 150 nm, resultant orders of magnitude are *D*_0_ ~ 1 × 10^−15^ cm^2^ s^−1^ and *Pe* ≥ 10^5^; this even exceeds the values characteristic of some colloids reported in strongly confined geometries and flows [48,49,50]. The colloidal systems are also entropy driven; metastable crystallization structures are determined by free-energy minima from a combination of electrostatic forces and interparticle interactions [51]. Other stabilization mechanisms are also possible, for example, sterically by ligands [52] (typical hard sphere case) or by electrostatic repulsion for charge stabilized particles [53]. The PO system, in comparison, does not have this inherent ability to self-assemble, and the equilibrium state is mainly generated by the accumulation and release mechanisms of strain energy from external macroscopic forces [54]. In such a system, where via short-range “sticky” interactions are prevalent, particles can additionally exert significant torque on each other, providing a mechanism by which excessive shearing causes shear melting and thus crystal dissolution. These are fundamental differences in behavior, meriting new theoretical approaches, offering critical insights into the microscopic mechanisms involved. 

In this paper, we present an experimental study of viscoelastic photonic crystal media, focusing on the key oscillatory shear-ordering dynamics. Direct measurement of the photonic stopband transmission in an optical shear cell allows the monotonic crystallization processes to be tracked as a function of oscillation frequency, strain amplitude, and temperature. A complementary generic model of crystallization due to shear-dependent interlayer viscosity is developed here, giving propagating crystalline fronts with increasing applied strain, and a gradual transition from disorder to order. The aim here is development towards a generic understanding of such systems, beyond the low strain/shear-rate assembly in, for example, colloids. The introduction into simulations of competing shear-induced flow degradation processes now give spatio-temporal solutions for the particle ordering, with both amplitude and frequency dependence. The extracted crystallization timescales, when degradation is dependent on the global shear rate, show parametric trends which are in close qualitative agreement with experimental observations.

## 2. Materials and Methods

### 2.1. Samples

The base core-interlayer-shell (CIS) particles for the polymer opal material described in this paper are illustrated in Figure 1. As previously reported, these are synthesized using a strategy of multistage emulsion polymerization [55,56,57]. The core-particle precursors used consist of a hard cross-linked polystyrene (PS) core, grown to approximately 230 nm in diameter, then coated with a thin (~10 nm) poly(methyl methacrylate) interlayer containing the comonomer allyl methacrylate (ALMA) as a grafting agent [58]. A softer polyethylacrylate (PEA) outer-shell was added, giving a total particle dimeter of ≈270 nm. Details of the particle size and dispersity characterization for the batch used in this report are given in Appendix A. The net refractive index contrast between core and shell material is thus Δ*n* ≈ 0.11 (or Δ*n*/*n* ≈ 7%), and the volume fraction of cores is ~55%. The CIS precursor batch in this case is modified by a 2.5% thiolation of the shell PEA material, facilitating rheological testing at a slightly increased Reynolds number (*Re*) than for many of the POs previously studied; the measured T_g_ value of ~−25 °C is some 10 °C lower than in earlier reports. The as-synthesized “polymer opal” is a stable viscoelastic quasi-solid, formed only as an ensemble of the composite particles with no separate solvent medium, and remained in this form during subsequent storage and use in the studies described here.

A general overview of the measured standard rheological parameters of the resultant PO material is given in Figure 1. The general signatures of viscoelastic behavior are confirmed, with the cross-over of the storage- and loss-modulus plots at strains of around 10–20%, indicating the yield point at which planes slippage may occur, in agreement with our earlier reports [31]. At a low oscillation frequency, sub-yield viscosities are in the range of 7000–8000 Pa·s at room temperature, decreasing to around 2000 Pa·s at 100 °C.

### 2.2. Shear Cell Assembly

The Linkam CSS450 shear cell [59] used in the experimental section of this work is illustrated in Figure 1. The as-synthesized PO composite material is carefully encapsulated between two parallel circular quartz windows with a radius of 1.5 cm and spacing set to 300 μm, giving a total sample volume of ~0.2 μL. Overlapping viewing ports with a radius of 1.4 mm are built into each supporting plate at a radius of ~1 cm from the center of the quartz windows, such as to allow continuous optical interrogation of the sample. Shearing of the sample is then achieved by a mechanical rotation of the bottom quartz window, relative to the fixed stationary top window. As the cell is of a cylindrical geometry, the shear motion of this arrangement thus provides only a close approximation to linear shear, with the strain defined at the point of observation in the (middle of the) observation window, which is offset from the center by a distance of 7.5 mm. To achieve the required range of viscoelastic response, the cell may be heated to thermocouple stabilized temperatures in the range of 20–100 °C.

The standard testing cycle used for samples in the shear cell is given in Table 1. In the initial step, a single continuous constant shear (2 s^−1^) over 10 s is employed to “randomize” the sample and establish a base condition of disorder, from which ordering could be initiated. Secondly, a 10 s period of stationary “relaxation” allowed any residual elastic forces present to dissipate. Finally, an oscillatory shearing step of up to 5 min is employed to “crystallize” the opal via shear ordering; by adjusting the transverse displacement of the cell plates (*δl*) over the sinusoidal cycle, the shear-strain amplitude (*σ*) was set by the ratio of *δl/d*, where *d* is the shear-cell sample thickness. Following the completion of the third step and the post-step relaxation of any residual phase-dependence, the sample can then be seen to gradually deteriorate away from the attained shear-ordered state over a period of some hours to days. This reiterates another distinction with shear-ordering colloidal systems, where a minimum shear rate is required to sustain stable crystallization and dissolution by diffusion would normally ensue.

As a general methodology, parametric variations in amplitude (*σ*), oscillation frequency (f) and temperature (T) are completed and reported as a discrete series of measurements, where the remaining two variables are fixed constants.

### 2.3. Microscopy/Spectroscopy

An adapted Olympus BX43 microscope, using an incandescent white light source focused to a measured spot size of approximately 10 μm in diameter (×5 magnification), is used to couple light through the sample cell. The transmitted light signal is then collected using suitable focusing optics and a fiber-coupled CCD spectrometer to enable real-time spectroscopic measurements. Spectra are taken from a tiny spot at the microscope focus in the middle of this frame, as verified by an independent measurement of light collection, across which the applied strain varies only by around 0.1%. Transmittance spectra are captured every 0.5 s during steps 1–3 of the experimental cycle and were then normalized against an appropriate control measurement of the empty cell. All the microscopic images displayed are taken with a 5 MP video camera and a standard RGB white-light balance. Data are primarily taken in transmission mode, as there are adverse practical issues of specular reflections from the optical windows and the normalization of spectra to overcome in reflectance. Secondly, the transmission mode yields information about light which has propagated across the complete 300 µm bulk thickness of the sample, whereas the reflectivity is only representative of sample properties down to the optical Bragg depth of order 10 µm.

As illustrated in Figure 2a, the key spectral dynamics are examined across the phases of the cycle by directly tracking the transmission/extinction coefficient at the center of the photonic stopband, λ = 610 nm. Tracking the short wavelength extinction, as per the methodology in earlier studies [32,46], relates only to the residual scattering (average density) from the disorder centers. Current experiments are unable to decouple residual background scatter coming from sources other than the periodic sphere structure, sphere size polydispersity or refractive index inhomogeneity, for example.

## 3. Results

### 3.1. Experimental Results

The progress of crystallization within the sample cell may also be followed visually in the microscope from the developing intense structural colors. Representative transmission micrograph images can be seen in in Figure 2a comparing an image frame capture where poor crystalline ordering is evident, with the intense green structural color following oscillatory shear. Due care is practiced in ensuring homogeneity within the areas from which spectra are subsequently gathered, as some small imperfections are evident over wider sample regions; these are mostly associated with localized cavities and contaminants, and edge effects.

In Figure 2b representative optical spectra taken during the phase of the testing cycle are shown during crystallization (oscillatory shear at 1 Hz, strain amplitude 150%). The progress of crystallization may be inferred from the change in transmission at the low-wavelength side of the resonance. Initially, there is a broad low transmittance across the spectrum, which is directly indicative of the particle spacing (radial distribution function) in the medium [60]. As ordering develops, the spectrum shows a marked stopband resonance centered around λ = 610 nm, with higher transmittance at both the low and high energy sides. The progressive red shifting of the resonance (≈ 580 to 610 nm) with increasing ordering, as the in-plane packing density increases, is also consistent with previous reports [16].

To gain a more satisfactory quantitative insight into the dynamics of crystal formation in the shear cell, the data are now plotted as extinction coefficients (at λ = 610 nm) against time. The on-resonance extinction coefficient (α) is here empirically defined as:(1)$$\alpha \;=\;\frac{-\ln\left(T\right)}{d},$$
where *T* is the fractional transmittance and *d* is the sample thickness. The clear monotonic temporal development of crystallization with oscillatory shearing is consistent with all of our earlier reports, whereby the increasing ordering was tracked spectroscopically using incremental multipass methods [16,17]. Intuitively, we expect there should therefore be an equilibrium point (with corresponding end-point coefficient, $\alpha_{0}$) at which the rate of defect generation is equal to the rate of crystallization. In this form, the data may be readily and conveniently fitted to an offset exponential function of the form:(2)α(t)= α0(1−Ae−tτ),
where $\alpha_{0}$ and *A* are offset constants and τ is the timescale. As a caveat, the experimental error in the measurement of *T* is ascertained to be in the order of 0.1%, which introduces a significant distributed baseline error in $\alpha_{0}$; this variation is not observed to be significant within a discrete series of measurements on the same sample load. However, changes in the residual scattering between different sample loads and experimental series do present such changes in the extinction baseline.

This generic timescale can also be usefully expressed in terms of the number of oscillation cycles (*N*) and frequency (*f*), that is:(3)$$\tau_{N}=\tau /f.$$

Additionally, *N* may be straightforwardly determined from the total cumulative applied strain (γ) and strain amplitude (σ) of the sinusoidal cycle, thus:(4)N=γ/4σ.

In Figure 3, the extinction coefficient time-dynamics are shown, for a series of different oscillation frequencies (fixed strain amplitude) and a series of strain amplitudes (constant frequency). In each case, shear-ordering timescales (and thus$\;\tau_{N}$) were extracted using exponential fitting as per Equation (2), and these values are shown in Table 2. Across the range of variables, some measurement cycles do not asymptotically reach $\alpha_{0}$ within 5 min (particularly at low frequency, low strain amplitude and at room temperature). Other samples reach $\alpha_{0}$ within a few seconds and indeed begin to exhibit some spectral changes and degradation before the end of step III, as the total applied strain at a high frequency and/or high strain amplitude begin to produce degradation by further motion.

Due to the finite increments of data sampling (0.5 s) relative to the oscillatory cycles, timescales of less than the order of a second are not practicably meaningful/attainable. Whilst the simple exponential fitting of data is therefore imperfect, the fit is reasonable within the reliable time window over which the ordering from base condition to equilibrium clearly proceeds, and gives us a predominantly characteristic timescale for the process.

#### 3.1.1. Strain Amplitude (σ) Dependence

In Figure 3a, representative α(t) plots are shown for strain amplitudes ranging from 50% to 300%, with the frequency and temperature fixed at 1 Hz and 50° C, respectively. As a clear trend, the ordering time/timescale is initially seen to decrease with increasing strain amplitude, with the shortest $\tau_{N}$ values occurring at around σ≈200%. As the amplitude is increased further, the ordering timescale is seen to increase again, being two orders of magnitude larger at σ≈300%; this is further analyzed in Table 2. As may also be inferred from Figure 3a, the end ranges of the amplitude (below σ≈100% and above σ≈250%) tend to show a suboptimal ordering efficacy, with a lower endpoint$\;\alpha_{0}$. Indeed, tests at 25% and 350% showed no measurable development of the photonic stopband within the timeframes of the experiment.

In agreement with earlier reports [16,17,30], it is seen that a larger strain is able to increase both the equilibrium level of ordering and also the speed of crystallization. A large shearing force on the polymer opal provides more stored elastic energy to facilitate sphere rearrangement with respect to a constant activation energy threshold. Intuitively, for effective crystallization to proceed, the stored energy must be greater than this required activation energy. However, very high strain values facilitate the dissipative processes competing with ordering, which are associated with shear-induced flow and displacement of particles; such processes also inhibit ordering of the opal from the base disordered condition [61].

#### 3.1.2. Oscillation Frequency (*f*) Dependence

In Figure 3b, representative α(t) plots are shown for oscillation frequencies ranging from 0.01 to 2.5 Hz (i.e., time periods of 100 down to 0.4 s), with strain amplitude and temperature fixed at 150% and 50 °C, respectively. The extracted ordering timescales must be explicitly converted into $\tau_{N}$ in this instance, according to the frequency dependence noted in Equation (3). Analysis shows that the shear ordering at lower frequencies is particularly effective; at 0.01 Hz, the value of $\tau_{N}$ is less than two complete cycles. At such low frequencies, the inherent hysteresis of the viscoelastic system can also be discerned in a way not readily possible at higher cyclical rates. Confirmation of the presence of this residual creep relaxation, superimposed upon the overall monotonic ordering trend, is consistent with earlier interpretations of the dependency and limitations of the ordering speed, where temporal asymmetry requires that strain rates are comparable to local creep rates in this highly viscous system.

As *f* is increased, $\tau_{N}\;$increases in a sublinear fashion, with around 20 completed cycles required for effective ordering above 2.5 Hz. However, the mechanical stability of the sample and equipment becomes less reliable at *f* = 5 Hz and higher. Making quantitative comparisons with the many experimental reports of shear crystallization in low *Pe* colloidal suspensions [37], crystallization was observed to proceed on timescales of 10 s of seconds to minutes, within comparable regimes of stress/strain (up to ~100%) and frequency (1–10 Hz) to those reported here. Generally, the relaxation behavior of these nonpermanent structures on the cessation of shear was markedly different.

#### 3.1.3. Temperature Dependence

Figure 3c, shows the temperature dependence of experimentally derived $\tau_{N}$ values for parameters of frequency = 1 Hz and strain = 100%, in comparison to the rheometrically measured viscosity. The ordering timescales decrease by approximately one order of magnitude in heating from room temperature up to 100 °C, with the overall trend tracking a corresponding softening of the PO. Rheological measurement indicates that storage (G’) and loss (G’’) cross-over, occurs at 40–50 °C, with the inferred viscosity illustrated in the figure for comparison, where there is commensurately the greatest rate of decrease in $\tau_{N}$(*T*).

### 3.2. Theory and Simulation

In this section, we expand earlier intuitive rheological ordering models, where order-dependent shear viscosities underpin the driving mechanisms in this athermal system (*Pe* ~ 10^5^–10^6^). These simulations gave an initial understanding of the exceptional ability of shear-ordering processes (such as BIOS) for inducing order in solvent-free viscoelastic systems of particles [62]. As the disordered spheres are compelled to separate into distinct planes, the shear viscosity drops, leading to a commensurate in-plane ordering where neighboring rows of spheres become aligned along the shearing direction. This is in broad agreement with the aforementioned microscopy studies on PO films, which revealed a layered cross-section with ordered regions propagating inwards from the outer surfaces. This is additionally, the pertinent laminar geometry in the experimental application of BIOS to the PET–PO–PET Timoshenko sandwich system [63].

However, whilst these previous models successfully show attributes strongly resembling the empirical observations in POs, such as a spatially broad crystallization front flowing from the interface and a saturated crystalline order behind it, there are areas for important further development. Critical parametric dependences, particularly those pertinent to oscillatory shear (amplitude, frequency etc.), were not encompassed. Nor were any competing order-degradation processes, such as shear melting, which would be anticipated especially at high strain amplitudes and frequencies, and which are clearly evident experimentally. In the following sections, we focus upon these necessary improvements to the efficacy and physical realism of our simulation models.

#### 3.2.1. Intuitive Model

Treating the PO as approximating a series of lamina layers stacked across the depth (*H*) of the sample, the natural units for spatial increments thus become 2*d*/*H*, where *d* is the sphere radius; successive layer elements are then defined by their index *i*. In order to parameterize the monotonic net ordering due to shear-induced thinning, a *local* shear rate
$\dot{\gamma }$
for the *i*^th^ layer is defined, which increases as the local crystalline order parameter *c*_i_ increases:(5)$${\dot{\gamma }}_{i}=a\cdot f\left(c_{i}\right),$$
where is proportional to the total shear strain applied across the entire film thickness. Parameter $c_{i}$ is a generic direction-averaged order parameter varying from zero (the disordered amorphous state) to one (optimized ordering condition). As per Figure 4d inset, we model the system as layers which when sheared, locally apply forces that lead to enhanced crystal order. The different types of order, which range from the stacking of planes to the nanoparticle ordering within each plane, are not distinguished in this model, and all lateral inhomogeneity is ignored. An increasing shear strain (as per when the optical cell windows are rotated with respect to each other) is nonuniformly distributed between different layers depending on their local order $c_{i}(z_{i}$,*t*) at depth $z_{i}$ and time *t*. According to Newton’s 2nd law, an additional velocity (α shear rate) added to each layer must generate an interlayer drag force that is proportional to the velocity difference between neighboring layers, that is:(6)$$F_{i}dt\propto \left(v_{i}-v_{i-1}\right)\propto \partial_{z}{\dot{\gamma }}_{i},$$

Since $v_{i}$ α${\dot{\;\gamma }}_{i}$, the incremental improvement in the local ordering, as described above, may be parameterized at each time point as,
(7)$$dc_{i}=u\cdot dt\left(v_{i}-v_{i-1}\right),$$
or generalizing to continuous spatio-temporal coordinates,
(8)$$\frac{\partial c}{\partial t}=u\frac{\partial \dot{\gamma }}{\partial z}=u\cdot a\left(t\right)\frac{\partial f}{\partial z},$$

The value of the constant u here is set following the semiempirical reports of Shereda et al. [64,65], as subsequently implemented by Zhao et al. [17]. By numerical integration, solutions, *c_i_ (z_i_,t)* may be obtained, for the development of ordering under applied shear across the sample depth *H*, with the normalization constraint on *a (t)* in terms of the total shear strain, $\gamma_{total}$, such that:(9)$$a\left(t\right)\int_{0}^{H}f\left[c\left(z,t\right)\right]dz=H\frac{d\gamma_{total}}{dt}.$$

A typical graphical output of $c_{i}(z_{i}$,*t*), showing the order parameter as a function of both applied strain and depth, can be seen in Figure 4a. Within this generic model, that incorporates the key elements of shear-induced ordering, the transfer function *f*(*c*) can be suitably adapted for different dependencies of the viscosity on the crystal ordering. In the case of a colloidal crystal, the transfer function is a simple “two-phase” step-function (Figure 4b), where only either amorphous or crystalline phases may exist. This produces the characteristic “growth front” behavior with a sharp threshold between ordered and amorphous regions [17,65]. However, for a viscoelastic medium characterized by a more gradual shear thinning and corresponding continuous monotonic transfer function, the model predicts a gradation of ordering across a wider front, as expected (Figure 4a). The progressive ordering in this system therefore also relies on irreversible dissipative forces acting upon errant spheres (Figure 4d inset); forces which are independent of the sign of the velocity difference between neighboring planes, and hence the oscillatory shear direction cycle.

To provide a better context of the spatial solutions generated, the value of $c_{i}(z$) is plotted as a function of shear strain (implicit *t* dependence) at depth points z = *H*/2 and *H*, in Figure 4c. These again illustrate the propagation of ordering, with c increasing monotonically at all points in space with the sample, before optimal ordering is asymptotically reached. To gain a point of comparison with the experimental interrogation methods, where the spectral signatures of ordering are probed across the shear-cell depth, a spatial averaging of c(z) is now introduced, for 0 < *z* ≤ *H* under standard simulation parameters,
strain amplitude (σ) = 150% and oscillation frequency (*f*) = 1 Hz. The spatially averaged order 〈c〉 is plotted against the number of oscillatory cycles (*N*) in Figure 4d, showing the exponential increase in sample ordering towards an asymptotic maximum value.

#### 3.2.2. Frequency and Amplitude Dependence

In steady-state shearing, the crystallites are pulled apart with increasing strain displacement. By contrast, oscillatory shear is able to cyclically nudge errant particles towards the lower viscosity (and more highly ordered) state, in strong corroboration with the experimentally observed behaviors. In order to account for the cyclicity of oscillatory shear, and the observed dependences on frequency and strain amplitude (rather than total linear strain), we now develop the basic intuitive model into two variants, both incorporating shear-induced flow degradation mechanisms.

In Model A, a simple competing degradation effect within each element is included by the addition of an extra term to Equation (8):(10)$${\dot{c}}_{i}=u\partial_{z}{\dot{\gamma }}_{i}-\kappa i,$$
where κ is a numerical constant, producing a linearly proportional degradation of the ordering in layer *i*. A representative *c_i_ (z_i_,t)* output from this model is shown in Figure 5a, together with the de facto time dependence of *c (z)* in Figure 5c and 〈*c*〉 as a function of *N* cycles in Figure 5d. The general behavior shows the expected gradual depth transition to ordering at lower shear strains, followed by a gradual disordering at a higher *N*; this is reminiscent of experimental observations in some PO samples, whereby a “wave” of the most optimally ordered regions is seen to migrate to below the interface with many repeated applications of BIOS. However, the spatio-temporal profile generated by this limited model shows unphysical behavior at a high *N*, as the degradation scales with the applied shear, and the system is broken up rather than equilibrating as intuition demands.

In Model B, we instead incorporate diametrically competing ordering and disordering terms for each element, and
${\dot{c}}_{i}$
is now dependent on the *global* shear rate (${\dot{\gamma }}_{g}$) across the depth of the film. Equation (8) can again be modified by the introduction of an additional linear term thus:(11)$${\dot{c}}_{i}=\left(u-k{\dot{\gamma }}_{g}\right)\partial_{z}{\dot{\gamma }}_{i}.$$

In this expression, k is a degradation constant with units of seconds, with u again being a constant associated with the shear-ordering rate. A representative $c_{i}(z_{i}$,*t*) output from this model is shown in Figure 5b, together with *c*(*z*) and 〈*c*〉 trends in comparison to Model A in Figure 5c,d. In addition to the intuitively correct monotonic ordering behaviors, whereby asymptotic ordering is reached at all depths at long times, the inherent dependence of the shear rate on both the oscillation frequency and strain amplitude now facilitates detailed quantitative analyses of these simulation outputs. A feature of this model is that a singularity is reached when the ordering and dissipative terms become equal, such that$\;{\dot{c}}_{i}\approx 0$ and therefore, $u=k{\dot{\gamma }}_{g}$. Taking the point at which net ordering ceases at a high strain amplitude, experimentally found to be at around σ ≈ 350%, and by evaluating the corresponding shear rate as being ${\dot{\gamma }}_{g}=4\sigma f$ = 14 s^−1^, it is possible to extract a value of k ≈ 0.5 s which is then used across all simulations to ensure physical consistency.

By applying a standard exponential best fit to the 〈*c*〉 functions of the form shown in Figure 5d for Model B, the simulated ordering timescales (i.e., τ and $\tau_{N}$) may be extracted, in a directly analogous fashion to the experimental data and the monotonic trend of Equation (2). A representative comparison is made in Figure 6a, showing the experimental time evolution of the stopband extinction coefficient along with the directly corresponding simulation outputs. This additionally reconfirms the preferred suitability of intuitive simulation model B over model A. In Figure 6b, the predicted values of $\tau_{N}$ are plotted as a function of the strain amplitude, with the experimental values from Table 2 in the range of σ = 50–350% also plotted for direct comparison. Particularly in the range of σ = 50–250%, there is strong quantitative agreement in the $\tau_{N}$ values and also a good qualitative correspondence between the overall functional trends. Only as the asymptotic ${\dot{c}}_{i}\approx 0$ point is approached at σ = 300–350% is there significant divergence between the experiment and simulation.

In Figure 6c, the predicted values of $\tau_{N}$ are plotted as a function of the oscillation frequency, with the corresponding experimental values from Table 2 in the range of *f* = 0.01 to 2.5 Hz plotted again for direct comparison. At low frequency, the two plots have both a good qualitative and quantitive agreement, with efficient development of ordering over few oscillatory cycles, and $\tau_{N}$ which increases gradually as a weak function of *f*. At above around *f* = 0.2 Hz, the qualitative agreement continues, as the number of required cycles for ordering increases monotonically. However, the model gives a rather steeper rate of rise of $\tau_{N}$ above around 1 Hz, whereas the experimental data shows a more gradual rise ($\tau_{N}$ increases approximately from 5 up to 15) in the range of 0.2 to 2 Hz.

## 4. Discussion and Conclusions

This study of thermal viscoelastic photonic crystal media in an oscillatory shear cell, confirms the monotonic nature of shear-ordering crystallization processes. The optically interrogated system dynamics were tracked experimentally as a function of key parameters (including strain amplitude, oscillation frequency, and temperature) over a far greater range than previously reported. This facilitates the quantitative improvement of complementary generic models, where shear-dependent interlayer viscosity reproduces the propagation of crystalline fronts and a gradual transition from disorder to order.

Spatio-temporal simulations with both strain amplitude and oscillation frequency dependence arise from the introduction of a competing shear-induced degradation process, dependent on the global shear rate. An encouraging correspondence between the experimental extracted and theoretically derived ordering timescales is evident from these “first-principles” intuitive models. These are important steps in the physical understanding and optimization of such engineered shear-ordering composite systems, including the polymer opal film arrays as bulk produced using analogous BIOS methods [27].

The simulated strain dependence of ordering efficacy and timescale shows a particularly good agreement with the experiment; confirming an optimal regime of shear amplitude in the range of σ = 100–250%, where local activation energies are exceeded, but below where flow degradation of the crystal is the dominant effect. The corresponding frequency dependence of the ordering timescale produces the same qualitative behavior between experimental data and simulation; on a cycle-by-cycle comparison, lower frequencies (*f* < 1 Hz) produce a more efficient ordering process, as shear-induced dissipation rates are low. Relative timescales then rapidly lengthen as *f* increases towards 1 Hz and higher.

Beyond the progress reported here, there are a number of ways in which better simulation convergence might be attempted, especially if a Monte Carlo or machine-learning approach becomes tractable. Firstly, the empirical constants relating to shear-dependent ordering and dissipation rates (u and k respectively in Equation (9)) may be fine-tuned within physical constraints. On a related theme, the model makes the ad hoc assumption of linearity between the crystal dissipation and global shear rate, as with the ordering forces. Whilst there is some intuitive justification for this, in the laminar flow regimes associated with low *Re*, the comparable suitability (or lack thereof) of nonlinear dissipation models may offer further physical insights. Moreover, the transfer function *f*(*c*) as applied in Equation (9), whilst highly successful in the initial implementation by Zhao et al., may be further tweaked to best represent the noncolloidal “sticky” interparticle interactions and gradation of ordering observed.

On a final note, a further challenge remains in the simulation of the temperature dependence of shear-ordering in this system. Whilst not straightforward or obvious from our earlier formalisms, the experimental correlation of ordering timescales with the bulk sample viscosity at least offers a future potential steer. It is wholly expected that, at such high characteristic Péclet numbers, the shear-dependent ordering and dissipation rates from our model of crystal formation will relate to rheology, and not to thermodynamics as for the colloidal suspension systems.

## Figures and Tables

**Figure 1 materials-14-05298-f001:**
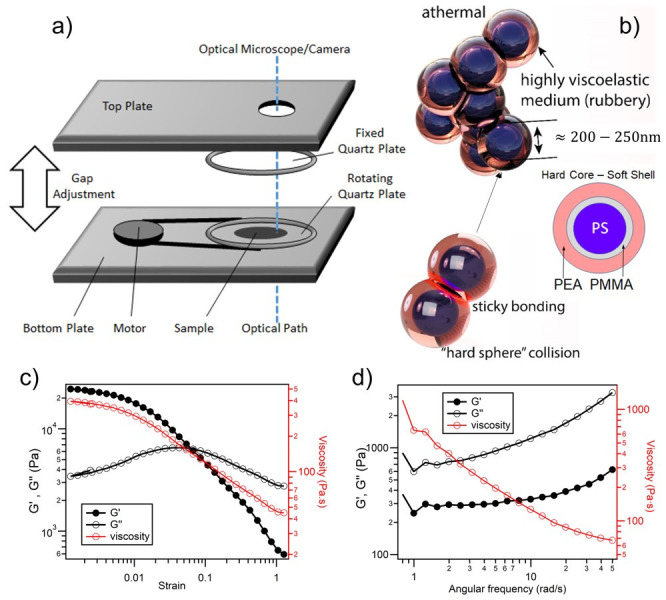
Schematic of an optical shear cell is shown in (**a**), with the optical path through the windows and cell, allowing real-time capture of transmission spectra, also illustrated. The composite core–shell structure, consisting of polystyrene (PS), poly-methylmethacrylate (PMMA) and poly-ethylacrylate (PEA) is shown in (**b**), together with a schematic of the ensemble interactions as particles form macroscopic polymer opal arrays. In (**c**,**d**), the measured storage (G’) and loss (G’’) moduli for the polymer opal material are given, together with the inferred viscosity. Strain dependence is shown in (**c**) at a 5 Hz frequency; angular frequency dependence is shown in (**d**) at 100% strain amplitude. Rheometric measurements are taken using an AR-2000 oscillatory rheometer (TA instruments), in a cone-and-plate geometry (radius 2 cm, angle = 1°, working gap = 27 mm). Image elements of part (**b**) have previously appeared under CC-BY license in reference [17].

**Figure 2 materials-14-05298-f002:**
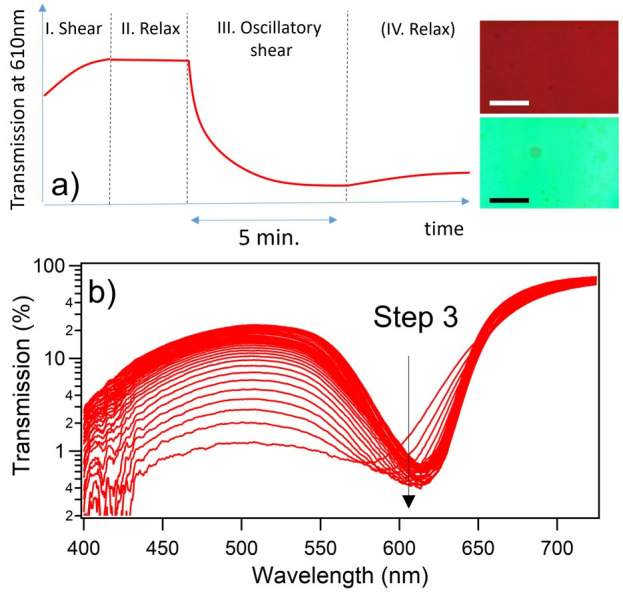
(**a**) Schematic illustration of the evolution of the stopband transmittance during the three-step characterization sequence, with a final relaxation step at the end of oscillatory shear. The photographic images in the inset compare the appearance of transmitted white light through the shear cell before (top) and after (bottom) the main oscillatory-shear cycle; uniform ordering, and a characteristically intense green structural color, are in evidence in the latter case. Both images were captured using an Olympus BX43 microscope, magnification ×5, and image scale bars ≈100 μm. In (**b**), transmission spectra are shown evolving with time during the oscillatory shear-ordering phase (step 3 in cycle). Successive spectra are 0.5 s apart in time, allowing a data slice at the photonic stopband minimum at λ = 610 nm, as illustrated by the arrow.

**Figure 3 materials-14-05298-f003:**
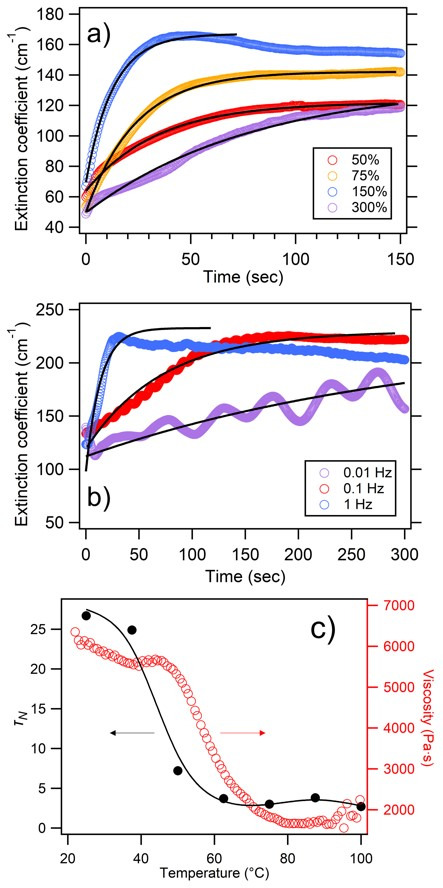
Time evolution of the extinction coefficient at the stopband wavelength of λ = 610 nm at a temperature of 50 °C. This is shown for various shear amplitudes at frequency 1 Hz in (**a**), and for various oscillation frequencies at a shear amplitude of 150% in (**b**), as indicated. Exponential trend lines are used to extract shear-ordering timescales, as limited to the illustrated range of fitting. (**c**) Shows the temperature dependence of experimentally derived $\tau_{N}$ values for parameters of frequency = 1 Hz and strain = 100%, in comparison to the rheometrically measured viscosity. An indicative trendline is added to the former, using smoothing spline fitting.

**Figure 4 materials-14-05298-f004:**
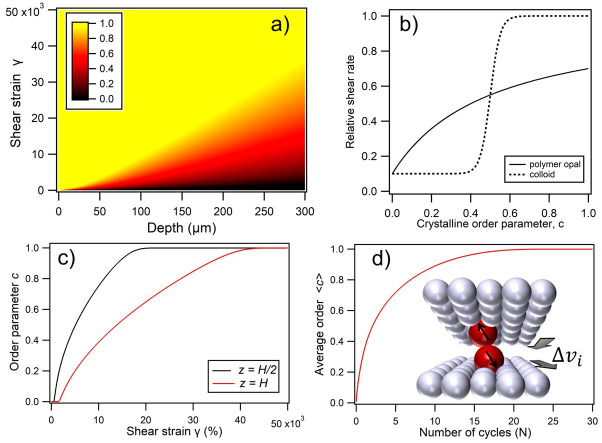
(**a**) Simulated nonlinear diffusion of crystal order into a sample with increasing total applied strain, showing a gradual depth transition which replicates experimental observations. The simulation parameters were strain amplitude = 150%, frequency = 1 Hz, and a z-scale for the ordering parameter, c, is shown as an inset. The nonlinear transfer relations between the normalized shear rate and crystalline order in each layer, *f*(*c*), used in simulations are plotted in (**b**), comparing the intuitive two-phase model for a colloid, with the polymer opal as indicated. (**c**) Shows how the order parameter varies at the cell midpoint (z = H/2) and full depth (H) as a function of total applied strain. (**d**) Shows how the order parameter, as spatially averaged across the 300 μm width of the sample, varies as a function of the number of BIOS oscillations (N). The inset schematic illustrates the assumed mechanism of ordering forces for individual errant spheres, where there is a relative layer velocity of Δν_i_.

**Figure 5 materials-14-05298-f005:**
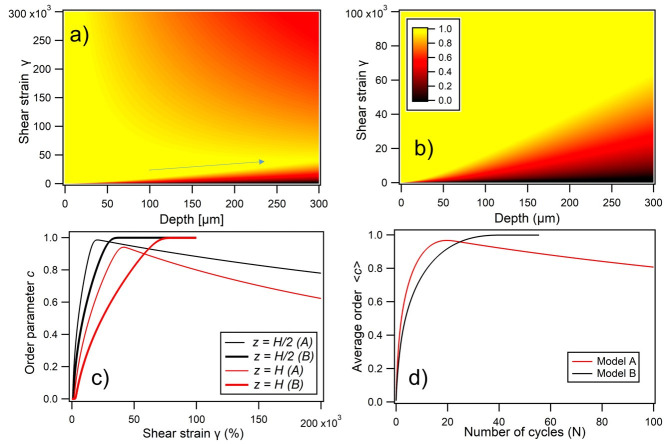
(**a**) Simulated nonlinear diffusion of crystal order into a sample with increasing total applied strain using Model A (degradation proportional to total applied strain), showing a gradual depth transition as illustrated by the arrow, followed by gradual disordering at high strains. The degradation constant κ has a notional value of 2.6 × 10^−10^ in this case. (**b**) Shows a comparative simulation using Model B (global shear-rate degradation), showing a gradual depth transition which replicates experimental observations. The degradation constant k has a value of 0.5 s in this case. In both (**a**,**b**), the simulation parameters were, strain amplitude = 150%, frequency = 1 Hz, and a common z-scale for the ordering parameter, *c*, is shown as an inset to (**b**). Comparing simulation models A and B, (**c**) shows how the order parameter varies at the cell midpoint (*z* = *H*/2) and full depth (*H*) as a function of total applied strain. Commensurately, (**d**) shows how the order parameter, as spatially averaged across the 300 mm width of the sample, varies as a function of the number of BIOS oscillations (N).

**Figure 6 materials-14-05298-f006:**
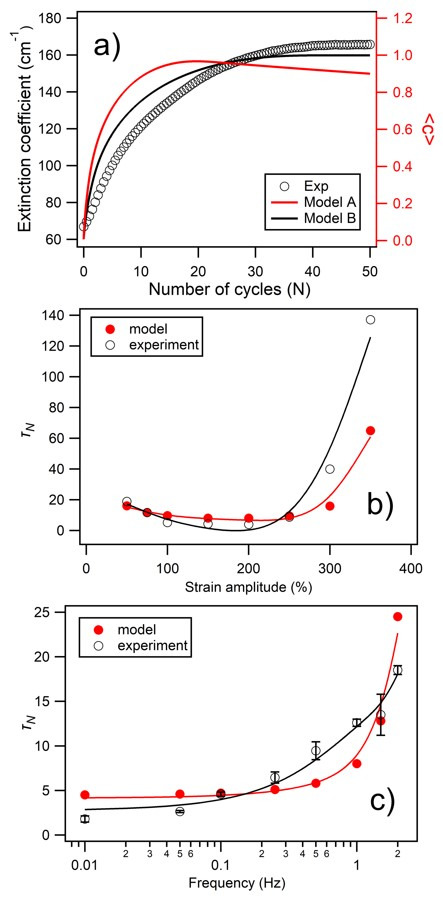
(**a**) Shows a representative side-by-side comparison of the experimental time evolution of the stopband extinction coefficient, with the directly corresponding simulation outputs for both Model A and Model B, in terms of the spatially averaged order parameter <c>. In this case, the temperature was 50 °C, frequency = 1 Hz, and shear amplitude = 150%. (**b**) Strain dependence of τ_N_ at 50 °C and frequency = 1 Hz, comparing the simulation model and experimentally derived values as indicated. (**c**) Shows the corresponding frequency dependence at strain amplitude 150%. In the lower two graphics, indicative trendlines are added, using smoothing spline fitting.

**Table 1 materials-14-05298-t001:** Standard three-step sequence used in the shear-cell characterization of opal samples. The mode of shearing, inter-plate gap, range of strain amplitude, range of shear rate, range of oscillation frequency and cycle time are shown for each step.

Step	Mode of Shear	Gap (μm)	Strain	Shear Rate (s^−1^)	Frequency (Hz)	Time (s)
1 (Randomize)	continuous	300	-	2	-	10
2 (Relaxation)	relaxation	300	-	-	-	10
3 (Crystallization)	oscillatory	300	25–350%	0.01–140	0.01–10	300

**Table 2 materials-14-05298-t002:** Extracted shear-ordering timescales (and thus $\tau_{N}$) as strain amplitude (σ), oscillation frequency (*f*) and temperature (*T*) are varied against the fixed values of σ = 100%, *f* = 1 Hz, and *T* = 50 °C. For the strain dependence data, the extracted extinction coefficients (*𝛼_0_*) at λ = 610 nm are also given.

Fixed *f*, *T*			Fixed *σ*, *T*			Fixed *f*, *σ*	
σ(%)	$\tau =\tau_{N\;}$	$\alpha_{0}$(cm^−1^)	f(Hz)	τ(s)	$\tau_{N}$	T(°C)	$\tau =\tau_{N}$
50	18.9 (±0.3)	121.4	0.01	178.7	1.8 (±0.4)	25	26.7 (±1.0)
75	11.6 (±0.1)	142.0	0.05	52.5	2.6 (±0.1)	37.5	24.9 (±1.1)
100	5.2 (±0.2)	184.4	0.1	45.8	4.6 (±0.2)	50	7.2 (±0.2)
150	4.3 (±0.1)	168.1	0.25	25.8	6.5 (±0.6)	62.5	3.7 (±0.1)
200	4.0 (±0.1)	139.8	0.5	18.9	9.5 (±1.0)	75	3.0 (±0.1)
250	8.7 (±0.2)	133.5	1.0	12.6	12.6 (±0.4)	87.5	3.8 (±0.3)
300	39.9 (±0.9)	113.8	1.5	9.0	13.5 (±2.3)	100	2.7 (±0.1)
350	137.1 (±8.9)	56.2	2.5	7.4	18.5 (±0.5)		

## Data Availability

Data supporting reported results available at doi: https://doi.org/10.20391/6c040db3-1368-4a97-8414-e3ec103ba5a0 (accessed date 3 September 2021).

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
