# Peer review of "An Experimental and Theoretical Determination of Oscillatory Shear-Induced Crystallization Processes in Viscoelastic Photonic Crystal Media"

_materials, 2021, doi:10.3390/ma14185298_

Round 1

Reviewer 1 Report

The manuscript by Finlayson et al. entitled “An experimental and theoretical determination of oscillatory 2 shear-induced crystallization processes in viscoelastic photonic 3 crystal media” deals with the shear-induced ordering for photonic crystals. This study follows a series of different studies by the same group. Here they introduce a simulation model that appears to describe the observed order formation qualitatively. While I find the data interesting and highly value the combined experimental and simulation approach, I have several concerns that need to be addressed before recommending the publication of the manuscript:

  1. Definition of Péclet number (line 87) is unclear to me in the discussed case. If no liquid medium is present, what is D0 and how is this determined? For colloidal systems, this is (mainly) given by the properties of the solvent, but here no solvent is present. E.g. with 150 nm particles in water at room temperature, one would obtain D0~10^-12 m^2/s and thus with gamma~1 s^-1: Pe~10^-2. The estimated diffusion constant can only be achieved with a very low viscosity. Please clarify the context. Note that Péclet number well above 10^3 and 10^4 can in principle be achieved for colloidal systems using microfluidic systems or liquid microjets as rheometers (see e.g. dois 10.1088/1367-2630/12/4/043056, 1073/pnas.1219340110, 10.1021/acs.jpclett.7b01355, 10.1063/4.0000038 for some examples).
  2. Lines 90-92: Colloidal systems are stabilized in different ways, e.g. sterically by ligands (typical hard sphere case) or by electrostatic repulsion for charge stabilized particles. Other stabilization mechanisms are also possible. Thus, “electrostatic forces” are also inter-particle interactions. In addition, there are better references than the cited thesis for this general statement.
  3. Section 3.1: What is the “core volume fraction”? Does it relate to on particle or to the total system? In the latter case, with a total diameter of 250 nm and 10 nm interlayer coating the total volume fraction would exceed 70 % which would only allow crystalline close packed structure – or deformed particles. Please clarify. Furthermore, how have these sizes been measured?
  4. How did you estimate the times needed for the different steps shown in Table 1? For instance, the transmission (shown in Fig. 2a) still increases the most towards the end of the first step. Can you be sure to have a full random structure and exclude e.g. small clusters?
  5. Comparing experiment and theory needs to be done on a similar basis.
    1. In the simulations, that parameter c ranges from 1 to 0, corresponding to “order” or “disorder”. What is actually meant quantitatively with “order” in this case? How does this connect, e.g., to Steinhardt parameters typically used in simulation and microscopy studies of order formation?
    2. In the experiment the order is given by the distinction coefficient. However, here no absolute scale is given! E.g. from Fig. 3 it is obvious, that not only the end values is different (and thus results in different “degrees of order”) but also the values at 0 seconds differ from each other. Consequently, I assume different values of A (Eq. 2) are obtained for each curve shown in Figure 3a and 3b and the comparability of the different times tau obtained from the modelling is questionable.
  6. To me the exponential models used for the data in Fig.3 a and b are too simple. First of all, the extinction coefficients seem to reach lower values at long times (e.g. 150% in Fig. 3a or 1 Hz in Fig. 3b) than anticipated from the models. This is mentioned by the authors, but I am missing an explanation or at least speculations about the origin. Keeping this observation out of the modelling – as done by the authors as well – the model is still far too simple. For me it looks like that at least two or more exponentials are needed to obtain a better trendline. Even for the yellow and red points in Fig. 3a this would provide likely much better matches, not speaking about other data points (worst the blue data in Fig. 3b). As the authors mentioned in the manuscript, this is an empirical model, so why not empirically assuming e.g. a two- or multi-step mechanism? Of course, this would result in different time scales that need to be addressed. In the current modelling, the error bars of the extracted times tau are surely too small. A logarithmic time scale would also help to see the actual development of the coefficients.
  7. In this context, simulation model B seems to model the trend of the data much better. However, the authors fit exponentials in the same way as for the experimental data and show the resulting tau’s in Fig. 6. Why not plotting the experimental data together with the model B for the order parameters? I do not understand that the authors may have found a way to model the data with a simulation, and instead doing so, they use a bad matching fit function to obtain quantitative parameters.
  8. The figures look a bit blurry to me. I recommend to increase the resolution – or maybe better – use a vector graphics format where possible (e.g. eps).

Minor:

  • Concerning the sample temperature, is this number given the actual temperature or the temperature of the sample cell? If so, did the author calibrate the temperature?
  • Instead of listing the data in Table, why not plotting them?
  • Check referencing. I think the journal guideline requests references before the full stop, e.g. “… Nature [1-4].” Instead of “… Nature. [1-4]”
  • Parts of Fig. 1 have been shown in Ref. 15, copyright may be needed.
  • Following the current IUPAC guidelines, use “dispersity” instead of “polydispersity” (line 49).
  • Use a tau symbol and the unit instead of “Tau-N” in the axis label of the figures.
  • I would prefer the SI unit “µm” instead of “microns”.
  • Maybe use a color scale in Figures 4a, 5a and 5b instead of grayscale, now differences are hardly visible.
  • Some typos need to be corrected, e.g. “stain” in the abstract, “pe” in line 213. In addition, some hyphens are missing and some sentences sound odd and could be simplified.
  • About 33% of the references are self-citations (some are not counted, e.g., the referenced theses). While here different philosophies exist, I found this number rather high.

Reviewer 2 Report

This research is devoted to shear-induced crystallization processes of photonic. Creating of nanostructured  materials is one of leading modern topic. The research is original and novel, however, the presentation of the results is presented in a confused way. Required correctiones are listed:

1) line 33-34, conventional strategies of making ordered materials should be listed (10.3390/nano10081538, 10.1103/PhysRevE.81.020401, 10.1063/1.1337619)

2) line 38-42, sentense is too big and confusing: if its highligting this paper, why there are refernce? Should reformulated

3) lines 43-47, the explanation of choosing objects it is not clear. Should be described why polystyrene spheres with core-shell structure was chosen and supported with references

4) line 79, information about shear-alignment of typical system - cellulose nanocrystals, and it's comparison with isotropic nanoparticles (polystyrene spheres and etc) is recommended to be added (10.1007/s10570-016-1150-4, 10.1002/adfm.202010743)

5) the quality of figure 1 should be improved

6) How was characterized core-shell spheres? Dynamical light scatterring results are recommended to be performed for prooving size and polydispersity

7) How exactly samples were prepared (weight concentration, pH, colloidal stability)?

8) quality of figure 2 should be improved. a),b) c) should be in one place of figure (now a) is in left side, b) and c) in right side). Description for used equipment for b) should be added

9) rheological data should from appendix 1 should be added at main part (not all, a, b or e)

10) line 374-378, using modeling from ref 15,54,55 should be clarified for this system (difference and approximations)

Round 2

Reviewer 1 Report

I appreciate the reply of the authors to the reviewer reports. Their answers are sufficient, and I can recommend publication. However, I still think that the modelling should be expanded as I stated in my first review - but I also acknowledge that this may go beyond the scope of the current manuscript and encourage to consider this for future work. I have only a small remark on the references, therefore I chose "minor revision": some references miss the correct page/article numbers (a common problem of some literature software), such as refs. 1, 17, 43, 48, 49, 62, 64 (may be more); the doi looks odd for refs. 6, 31; ref. 51 appears in capitals only. Please check and correct.

Author Response

We are very thankful to the reviewer for their attention and appraisal of our manuscript.

As requested, the reference formatting has now been re-checked, and corrected where appropriate. We hope this now makes the manuscript suitable for publication as an invited contribution to MDPI Materials.